# Role of a Real-Time TDM-Based Expert Clinical Pharmacological Advice Program in Optimizing the Early Pharmacokinetic/Pharmacodynamic Target Attainment of Continuous Infusion Beta-Lactams among Orthotopic Liver Transplant Recipients with Documented or Suspected Gram-Negative Infections

**DOI:** 10.3390/antibiotics12111599

**Published:** 2023-11-07

**Authors:** Milo Gatti, Matteo Rinaldi, Cristiana Laici, Antonio Siniscalchi, Pierluigi Viale, Federico Pea

**Affiliations:** 1Department of Medical and Surgical Sciences, Alma Mater Studiorum University of Bologna, 40138 Bologna, Italy; mat.rinaldi1989@gmail.com (M.R.); pierluigi.viale@unibo.it (P.V.); federico.pea@unibo.it (F.P.); 2Clinical Pharmacology Unit, Department for Integrated Infectious Risk Management, IRCCS Azienda Ospedaliero-Universitaria of Bologna, 40138 Bologna, Italy; 3Infectious Disease Unit, Department for integrated Infectious Risk Management, IRCCS Azienda Ospedaliero-Universitaria of Bologna, 40138 Bologna, Italy; 4Anesthesia and Intensive Care Medicine, IRCCS Azienda Ospedaliero-Universitaria di Bologna, 40138 Bologna, Italy; cristiana.laici@aosp.bo.it (C.L.); antonio.siniscalchi@aosp.bo.it (A.S.)

**Keywords:** orthotopic liver transplant recipients, beta-lactams, continuous infusion, intensive care unit, Gram-negative infections, PK/PD target attainment, expert clinical pharmacological advice, augmented renal clearance

## Abstract

(1) Objectives: To describe the attainment of optimal pharmacokinetic/pharmacodynamic (PK/PD) targets in orthotopic liver transplant (OLT) recipients treated with continuous infusion (CI) beta-lactams optimized using a real-time therapeutic drug monitoring (TDM)-guided expert clinical pharmacological advice (ECPA) program during the early post-surgical period. (2) Methods: OLT recipients admitted to the post-transplant intensive care unit over the period of July 2021–September 2023, receiving empirical or targeted therapy with CI meropenem, piperacillin-tazobactam, meropenem-vaborbactam, or ceftazidime-avibactam optimized using a real-time TDM-guided ECPA program, were retrospectively retrieved. Steady-state beta-lactam (BL) and/or beta-lactamase inhibitor (BLI) plasma concentrations (C_ss_) were measured, and the C_ss_/MIC ratio was selected as the best PK/PD target for beta-lactam efficacy. The PK/PD target of meropenem was defined as being optimal when attaining a *f*C_ss_/MIC ratio > 4. The joint PK/PD target of the BL/BLI combinations (namely piperacillin-tazobactam, ceftazidime-avibactam, and meropenem-vaborbactam) was defined as being optimal when the *f*C_ss_/MIC ratio > 4 of the BL and the *f*C_ss_/target concentration (C_T_) ratio > 1 of tazobactam or avibactam, or the *f*AUC/C_T_ ratio > 24 of vaborbactam were simultaneously attained. Multivariate logistic regression analysis was performed for testing potential variables that were associated with a failure in attaining early (i.e., at first TDM assessment) optimal PK/PD targets. (3) Results: Overall, 77 critically ill OLT recipients (median age, 57 years; male, 63.6%; median MELD score at transplantation, 17 points) receiving a total of 100 beta-lactam treatment courses, were included. Beta-lactam therapy was targeted in 43% of cases. Beta-lactam dosing adjustments were provided in 76 out of 100 first TDM assessments (76.0%; 69.0% decreases and 7.0% increases), and overall, in 134 out of 245 total ECPAs (54.7%). Optimal PK/PD target was attained early in 88% of treatment courses, and throughout beta-lactam therapy in 89% of cases. Augmented renal clearance (ARC; OR 7.64; 95%CI 1.32–44.13) and MIC values above the EUCAST clinical breakpoint (OR 91.55; 95%CI 7.12–1177.12) emerged as independent predictors of failure in attaining early optimal beta-lactam PK/PD targets. (4) Conclusion: A real-time TDM-guided ECPA program allowed for the attainment of optimal beta-lactam PK/PD targets in approximately 90% of critically ill OLT recipients treated with CI beta-lactams during the early post-transplant period. OLT recipients having ARC or being affected by pathogens with MIC values above the EUCAST clinical breakpoint were at high risk for failure in attaining early optimal beta-lactam PK/PD targets. Larger prospective studies are warranted for confirming our findings.

## 1. Introduction

Orthotopic liver transplant (OLT) is the most effective strategy for dealing with end-stage liver failure caused by cirrhosis [1]. Despite a significant improvement in surgical techniques and immunosuppressant regimens, and a prompt identification of post-transplant complications that has been obtained over the last few years, bacterial infections still represent the predominant cause of post-OLT morbidity and mortality [1]. Furthermore, donation after circulatory death (DCD) has recently emerged as a relevant way for bridging the gap between donor liver graft availability and the number of patients waiting for an OLT [2]. Unfortunately, recent studies have reported that the infection rate after DCD OLT was remarkable, involving approximatively 70% of cases [3]. Bloodstream infections (BSIs) and ventilator-associated pneumonia (VAP) caused by Gram-negative pathogens represent the most frequent types of infection in early post-OLT surgery during intensive care unit (ICU) stays [4,5,6,7,8,9,10,11].

Beta-lactams currently represent a backbone therapy for managing Gram-negative infections in solid organ transplant (SOT) recipients, including OLT [12], and they are the first-line prophylaxis of surgical site infections in OLT recipients [13]. In this scenario, the prompt optimization of beta-lactam exposure may play a key role, given the remarkable pathophysiological alterations that have been commonly observed in critically ill patients, potentially leading to a high risk of changeable exposure [14]. Real-time therapeutic drug monitoring (TDM)-based expert clinical pharmacological advice (ECPA) programs may be helpful for optimizing beta-lactam exposure in OLT recipients according to the so-called antimicrobial therapy puzzle concepts [15].

Recently, the attainment of an aggressive pharmacokinetic/pharmacodynamic (PK/PD) target of 100%*f*T_> 4–8 × MIC_ with the continuous infusion (CI) of beta-lactams was associated with both the maximization of clinical efficacy and the suppression of resistance development in critically ill patients [16,17,18]. In a recent study, it was found that among 70 lung transplant recipients, beta-lactam PK/PD target attainment in the early post-transplant phase was suboptimal in as much as 40% of cases, and this led to significantly higher risks of multidrug-resistant (MDR) Gram-negative colonization and infections [19].

The aims of this study were, on the one hand, to assess the role that a real-time TDM-based ECPA program may have had in enabling the prompt attainment of optimal PK/PD targets of CI beta-lactams in a population of critically ill OLT recipients undergoing the empirical or targeted therapy of early onset post-transplant Gram-negative infections, and, on the other hand, to identify the potential independent predictors of suboptimal/quasi-optimal PK/PD target attainment.

## 2. Results

Overall, 255 OLT recipients were admitted to the post-transplant ICU during the study period. Among them, 77 patients had CI beta-lactams exposure personalized through TDM-based ECPAs in the early post-transplant period and these were included in the study (Figure 1). The demographics and clinical features of the patients are summarized in Table 1.

The median [interquartile range (IQR)] age was 57 years (51–63 years), with a male preponderance (63.6%). The Median (IQR) Model for End-Stage Liver Disease was 17 (11–29). In seven cases (9.1%), DCD was implemented. Primary sclerosing cholangitis (12 cases; 15.5%), alcoholic plus dysmetabolic cirrhosis (nine cases; 11.7%), alcoholic cirrhosis (seven cases; 9.1%), and HCV + hepatocarcinoma (seven cases; 9.1%) were the most frequent underlying liver diseases. Five patients (6.5%) underwent re-OLT because of primary non-function (four cases) or chronic rejection (one case).

At ICU admission, the median (IQR) Sequential Organ Failure Assessment (SOFA) score was 6.5 (3.75–9.25). Thirty-six patients (46.8%) underwent invasive mechanical ventilation for at least 48 h, and 50 (64.9%) required cardiovascular support with vasopressors. Continuous renal replacement therapy (CRRT) was applied in 28 cases (36.4%), and augmented renal clearance was documented in 15 cases (19.5%). None of the included patients received probiotic supplements. The ICU mortality rate was 9.1%.

Overall, 245 TDM-based ECPAs were performed for optimizing 100 beta-lactam treatment courses among the 77 OLT recipients (Table 2).

Beta-lactam therapy was empirical in 57 courses (57.0%) and was targeted in the other 43 (43.0%). Meropenem, piperacillin-tazobactam, meropenem-vaborbactam, and ceftazidime-avibactam were used in 45, 44, 7, and 4 treatment courses, respectively.

Infection types were ventilator-associated pneumonia (VAP) in 17/43 cases (39.5%), complicated intrabdominal infection (cIAI) in 11/43 cases (25.6%), bloodstream infection (BSI) in 9/43 cases (20.9%), cIAI plus BSI in 4/43 cases (9.3%), and VAP plus BSI in two cases (4.7%). Overall, 51 different Gram-negative pathogens were isolated, with *Klebsiella pneumoniae* (31.3%), *Enterobacter cloacae* (15.7%), *Escherichia coli* (13.7%), and *Pseudomonas aeruginosa* (13.7%) being the most frequent ones. ESBL-, AmpC-, and/or carbapenemase-producers accounted for 29 out of 41 *Enterobacterales* clinical isolates (70.7%).

A total of 245 TDM-guided ECPAs were performed, with a median (IQR) of 2 (1–3) per treatment course. Dosing adjustments at first TDM-guided ECPA were performed in 76 out of 100 cases (76.0%), with 69 decreases (69.0%) and 7 increases (7.0%), respectively. Overall, beta-lactam dosing adjustments were recommended in 54.7% of TDM-guided ECPAs (48.6% decreases and 6.1% increases).

PK/PD target attainments of each beta-lactam are summarized in Table 3.

Overall, the optimal PK/PD target was attained in 126 out of 145 TDM-guided ECPAs provided for meropenem (89.4%), in 69 out of 79 of those provided for piperacillin/tazobactam (87.3%), in 6 out of 11 of those provided for meropenem-vaborbactam (54.5%), and in 13 out of 14 (92.9%) of those provided for ceftazidime-avibactam. At the first TDM assessment, the early attainment of the optimal PK/PD target was observed in 40/45 (88.9%) of meropenem courses, in 40/44 of piperacillin-tazobactam courses (90.9%), in 5/7 of the meropenem-vaborbactam courses (71.4%), and in 3/4 ceftazidime-avibactam courses (75.0%) (Figure 2).

Univariate and multivariate regression analyses testing variables possibly associated with suboptimal PK/PD target attainment are shown in Table 4.

Overall, at multivariate analysis, ARC (odds ratio [OR] 7.64; 95% confidence interval [CI] 1.32–44.13; *p* = 0.023) and MIC value > EUCAST clinical breakpoint (OR 91.55; 95%CI 7.12–1177.12) emerged as independent predictors of the suboptimal/quasi-optimal early PK/PD target attainment of beta-lactams.

## 3. Discussion

To the best of our knowledge, this is the first study that has assessed the role of a real-time TDM-based ECPA program in enabling the prompt attainment of optimal PK/PD targets of CI beta-lactams in a population of critically ill OLT recipients undergoing the empirical or targeted therapy of early onset post-transplant Gram-negative infections. Our findings suggested that the program allowed for prompt optimal PK/PD target attainment in approximately 90% of cases throughout treatment courses with CI meropenem, piperacillin-tazobactam, meropenem-vaborbactam, and ceftazidime-avibactam. Additionally, ARC and MIC values against the clinical isolate above the EUCAST clinical breakpoint have emerged as independent predictors of suboptimal/quasi-optimal PK/PD target attainment.

Several studies have shown that Gram-negative infections have a major role during the early post-OLT period, with pneumonia, bacteremia, and intrabdominal infections being the most frequent ones [1,4,5,6,7,8,9,10,11,20,21]. In this scenario, beta-lactams are first-line treatments [12], and attaining an optimal PK/PD target early may allow for the maximization of clinical efficacy and suppress resistance development with beta-lactams [17,18]. Unfortunately, the complex pathophysiological conditions of OLT patients may deeply alter the pharmacokinetic behavior of hydrophilic agents like the beta-lactams, thus hampering the possibility of attaining adequate PK/PD targets [14,22,23,24,25,26]. In this regard, a TDM-guided ECPA program may be found to be very useful [27]. Our real-time TDM-guided ECPA program found that only approximately 10% of OLT recipients receiving CI beta-lactams failed in attaining early optimal PK/PD targets. This finding is in disagreement with a previous study showing that the intermittent infusion of cefepime, piperacillin-tazobactam, and meropenem resulted in a 40% suboptimal target attainment among 70 ICU-admitted lung transplant recipients [19]. This may be explained by taking into account that the adoption of CI administration, as we have always done, may be a very valuable strategy for maximizing the likelihood of attaining aggressive PK/PD targets with beta-lactams under the same daily dose, as recently reported [28,29]. In this regard, the TDM-guided ECPA program, by enabling the prompt identification of the minority of patients with eventual suboptimal exposure, may be effective in granting optimal PK/PD target attainment with beta-lactams in the whole patient population, which is different from standard approaches [30,31].

The finding of ARC as an independent risk factor of early suboptimal and/or quasi-optimal PK/PD target attainment is in agreement with several studies showing a significant association between ARC and a failure in attaining optimal PK/PD targets with beta-lactams, namely, an occurrence that may possibly lead to worse clinical outcomes [19,32,33,34,35,36,37,38,39,40]. This is not surprising, considering that beta-lactams are predominantly renally cleared [28,41], and this should push clinicians to adopt more intensified dosing regimens for properly treating OLT recipients having ARC with beta-lactams [14].

Also, MIC values against Gram-negative bacterial isolates, being above the EUCAST clinical breakpoint, emerged as an independent predictor of failure in attaining early optimal PK/PD targets among OLT recipients treated with beta-lactams. A recent study conducted among 21 critically ill pediatric patients showed that quasi-optimal/suboptimal beta-lactam PK/PD target attainment occurred more frequently among patients having infections caused by less susceptible pathogens with borderline in vitro susceptibility [42]. Clearly, this is in agreement with the fact that the beta-lactam doses for properly dealing with this issue should be higher than the standard ones [43]. In this regard, our study showed that administering high beta-lactam dosing regimens via CI and optimizing PK/PD target attainment by means of a TDM-guided ECPA program was a valuable strategy for maximizing treatment efficacy, even in this challenging scenario.

Another major finding was the fact that at the first TDM assessment, the TDM-based ECPA program allowed for the reduction of CI beta-lactam dosing regimens in about 70% of patients, thus potentially preventing the risk of prolonged overexposure and toxicity in patients with persisting multiorgan failure [44,45]. Conversely, in subsequent TDM-guided ECPA reassessments, dosing regimens were confirmed in up to 60% of cases, thus supporting the potentially relevant role of this approach in personalizing beta-lactam treatments among critically ill OLT recipients.

Limitations of our study have to be recognized. The retrospective monocentric study design must be acknowledged. Total beta-lactam concentrations were measured, and the free fractions were only estimated based on the percentage of plasma protein binding retrieved in healthy volunteers. Conversely, the fact that this is the first real-life experience describing the PK/PD target attainment of beta-lactams in critically ill OLT recipients during the early post-transplant period may be considered a point of strength of our study.

## 4. Materials and Methods

### 4.1. Study Design

OLT recipients who were admitted at the post-transplant ICU of the IRCCS Azienda Ospedaliero-Universitaria of Bologna, Italy in the period between 1 July 2021 and 15 September 2023 were retrospectively analyzed for possible inclusion in this study. The inclusion criteria were: (a) empirical or targeted therapy with CI beta-lactams, namely, piperacillin-tazobactam, meropenem, ceftazidime-avibactam, or meropenem-vaborbactam during the early post-surgery ICU period; (b) the optimization of PK/PD target attainment of these beta-lactams by means of a real-time TDM-guided ECPA program. Early post-OLT admission in ICU was defined as the 30-day post-transplant period and included either immediate post-OLT admission or subsequent re-admissions because of complications [4,46]. Patients having TDM-guided ECPAs for beta-lactam treatment outside of the post-transplant ICU or after more than 30 days from OLT were excluded.

### 4.2. Data Collection

Demographic (age, sex weight, height, and body mass index (BMI)) and clinical/laboratory data (underlying disease leading to OLT, pre-OLT Model for End-Stage Liver Disease (MELD) score, performance of a combined liver–kidney transplantation and/or of a DCD OLT and SOFA score at ICU admission, the need for mechanical ventilation and/or for vasopressor support, requirement for CRRT, and occurrence of ARC) were collected for each patient. ARC was defined as a measured urinary creatinine clearance (based on 24 h urine collection) above 130 mL/min and 120 mL/min in males and females, respectively. Beta-lactam dosing and average plasma steady-state concentrations (C_ss_), overall number of ECPAs, ECPA-recommended dosing adjustments at first and at subsequent TDM assessment, and ICU mortality rate were also retrieved. In the case of targeted therapy, clinical isolates, MIC values of beta-lactams against specific clinical isolate, and type/site of infection were collected. The MIC values for piperacillin-tazobactam and meropenem against Gram-negative clinical isolates (*Enterobacterales*, *Pseudomonas aeruginosa*, and/or *Acinetobacter baumannii*) were measured by means of a semi-automated broth microdilution method (Microscan Beckman NMDRM1), whereas those for ceftazidime-avibactam and meropenem-vaborbactam against carbapenem-resistant *Enterobacterales* were tested according to a broth microdilution method and interpreted according to the European Committee on Antimicrobial Susceptibility Testing (EUCAST) clinical breakpoints [47]. MIC values ≤ 2 mg/L for meropenem and ≤8 mg/L for piperacillin-tazobactam, ceftazidime-avibactam, and meropenem-vaborbactam identified susceptible pathogens.

Centers for Disease Control and Prevention (CDC) criteria were used for defining the different types of infection [48]. Specifically, documented BSI was defined as the isolation of a Gram-negative pathogen from at least one blood culture [48]. Documented VAP was defined as the isolation of one or more Gram-negative pathogens with a bacterial load ≥10^4^ CFU/mL in the bronchoalveolar lavage (BAL) fluid culture after > 48 h from endotracheal intubation and the initiation of mechanical ventilation in patients showing new or progressive lung infiltrates [49,50]. cIAI was defined as the isolation of one or more Gram-negative pathogens from the peritoneal fluid or abdominal specimens [50,51].

### 4.3. Beta-Lactam Dosing Regimens, Sampling Procedure, and Procedure for Optimizing PK/PD Target Attainment

Empirical or targeted treatment with beta-lactams was prescribed by the treating intensive care physicians and/or the infectious disease consultant according to the underlying conditions of each patient and the results of antimicrobial susceptibility tests. For each selected beta-lactam, treatment was started with a loading dose (LD) (namely, 9 g, 2 g, 2.5 g, and 2 g/2 g, administered over 2 h infusion for piperacillin-tazobactam, meropenem, ceftazidime-avibactam, and meropenem-vaborbactam, respectively) followed by an initial maintenance dose (MD) administered through CI. According to different stability restrictions, aqueous solutions were reconstituted every 6–8 h and infused over 6–8 h for meropenem and/or meropenem-vaborbactam [52,53,54], every 8–12 h and infused over 8–12 h for ceftazidime-avibactam [52], and every 24 h and infused over 24 h for piperacillin-tazobactam [52].

Initial MD regimens were defined on a case-by-case basis, taking into account the patient’s underlying conditions and renal function, the site of infection, and the presence/absence of a bacterial isolate with the corresponding MIC value. The dosing was subsequently optimized by means of a real-time TDM-guided ECPA program. For this purpose, blood samples were collected firstly at least 24 h from the starting therapy for measuring beta-lactams C_ss_, and then reassessed every 48–72 h whenever feasible. Total piperacillin-tazobactam, meropenem, ceftazidime-avibactam, and meropenem-vaborbactam plasma concentrations were determined by means of validated liquid chromatography-tandem mass spectrometry methods [17,55,56].

Real-time TDM-guided ECPAs were provided by well-trained MD clinical pharmacologists (ECPA) who attended Monday-to-Friday morning bedside multidisciplinary meetings in the ICU. The ECPA was structured by taking into account some specific underlying conditions, including the in vitro susceptibility of the pathogens, the site of infection, and the patient’s pathophysiological conditions [15].

### 4.4. Definition of Optimal, Quasi-Optimal, and Suboptimal PK/PD Target Attainments of Beta-Lactams

The percentage of time with free beta-lactam C_ss_ above the MIC was selected as the best PK/PD determinant of beta-lactam efficacy and expressed as the *f*C_ss_/MIC ratio (equivalent to %*f*T_> MIC_).

Aggressive PK/PD targets were selected based on preclinical and clinical studies reporting that the attainment of these targets may be associated with both the maximization of clinical efficacy and the suppression of resistance emergence against Gram-negative pathogens with beta-lactams [16,17,18,57]. PK/PD targets were arbitrarily defined as optimal, quasi-optimal, and suboptimal, according to the following rules. With regard to meropenem, PK/PD target attainment was defined as being optimal when the *f*C_ss_/MIC ratio was >4 (equivalent to 100%*f*T _> 4 × MIC_), and quasi-optimal or suboptimal when the *f*C_ss_/MIC ratio was 1–4 or <1 (equivalent to 100%*f*T_1–4 × MIC_ and to <100%*f*T_1 × MIC_), respectively, as previously reported [58]. With regard to the BL/BLI combinations, namely piperacillin-tazobactam, ceftazidime-avibactam, and meropenem-vaborbactam, a joint PK/PD target was considered [59,60]. With regard to piperacillin-tazobactam, the joint PK/PD target was defined as being optimal if both the piperacillin *f*C_ss_/MIC ratio was >4 and the tazobactam *f*C_ss_/target concentration (C_T_) ratio was >1, and quasi-optimal or suboptimal if only one or none of the two thresholds was attained, respectively [59]. With regard to ceftazidime-avibactam, the joint PK/PD target was defined as being optimal if both the ceftazidime *f*C_ss_/MIC ratio was >4 and the avibactam *f*C_ss_/C_T_ ratio was >1, and quasi-optimal or suboptimal if only one or none of the two thresholds was achieved, respectively [60]. With regard to meropenem-vaborbactam, the joint PK/PD target was defined as being optimal if both the meropenem *f*C_ss_/MIC ratio was >4 and the vaborbactam free area under the concentration-to-time curve (*f*AUC)/C_T_ ratio was >24, and quasi-optimal or suboptimal if only one or none of the two thresholds was attained, respectively [61]. The AUC of vaborbactam was calculated by means of the following formula: AUC (mg∙h/L) = dose (mg/24 h)/clearance [CL] (L/h), where CL was equal to the infusion rate (mg/h)/C_ss_ (mg/L). The C_T_ corresponded to the fixed BLI target concentration used by the EUCAST for the in vitro standard susceptibility testing of each of the BL/BLI combinations, namely, 4 mg/L for tazobactam and avibactam, and 8 mg/L for vaborbactam.

The free fractions of beta-lactams and beta-lactamase inhibitors were calculated according to the protein binding rate reported in the literature, namely, 2% for meropenem [62], 20% for piperacillin [63], 23% for tazobactam [63], 10% for ceftazidime [64], 7% for avibactam [64], and 33% for vaborbactam [65].

Beta-lactam dosing adjustments were performed as previously reported [15]. Briefly, for beta-lactams, a 25% or 50% dosing decrease was adopted whenever the *f*C_ss_/MIC ratio was equal to 8–10 or >10, respectively; dosing was confirmed whenever the *f*C_ss_/MIC ratio was equal to 4–8; and a 25% or 50% dosing increase was implemented whenever the *f*C_ss_/MIC ratio was equal to 2–4 or below 2, respectively. For the BL/BLI combinations, the dosing increase was also implemented when the tazobactam or avibactam *f*C_ss_/C_T_ ratio was <1, or when the vaborbactam *f*AUC/C_T_ ratio was <24, as previously reported [59,60].

In patients undergoing more than one TDM-guided ECPAs, the average BL and BLI C_ss_ were considered by calculating the mean of all observed C_ss_ values (the first one before any dosage adjustment and the subsequent ones after eventual dosage adjustments). For each treatment course, the attainment of the optimal beta-lactam PK/PD target was assessed both first (i.e., early PK/PD target attainment) and through subsequent TDM-guided ECPA assessment (i.e., by considering all of the delivered TDM-guided ECPAs). As for the MIC value, it was considered the EUCAST clinical breakpoint against the suspected pathogen in the case of empirical treatment (namely, 2 mg/L for meropenem and 8 mg/L for piperacillin-tazobactam, ceftazidime-avibactam, or meropenem-vaborbactam), and the punctual MIC value of the clinical isolate in the case of targeted therapy, as previously defined [15].

### 4.5. Statistical Analysis

Continuous data were expressed as median and interquartile range (IQR), and categorical variables were presented as counts or percentages. Univariate analysis between beta-lactam treatment courses attaining an early optimal vs. quasi-optimal/suboptimal PK/PD target was performed by means of Fisher’s exact test or the chi-squared test (for categorical variables), or the Mann-Whitney U test (for continuous variables). Multivariate logistic regression analysis was implemented for testing possible variables associated with a failure in attaining early optimal PK/PD targets. Independent covariates with a *p* value < 0.10 in the univariate analysis were included in the multivariate logistic regression model. Statistical significance was defined as a *p* value < 0.05. Statistical analysis was performed by using MedCalc for Windows (MedCalc statistical software, version 19.6.1, MedCalc Software Ltd., Ostend, Belgium).

## 5. Conclusions

Overall, our findings indicated that a real-time TDM-based ECPA program allowed for optimal PK/PD target attainment in approximately 90% of critically ill OLT recipients treated with CI beta-lactams during the early post-transplant period. The findings of ARC and/or of MIC value against the clinical isolates above the EUCAST clinical breakpoint as independent predictors of only suboptimal/quasi-optimal PK/PD target attainment should push clinicians to adopt more intensified beta-lactam dosing regimens for granting optimal PK/PD target attainment whenever they are dealing with OLT recipients having these challenging conditions. Larger prospective studies are warranted for confirming our findings.

## Figures and Tables

**Figure 1 antibiotics-12-01599-f001:**
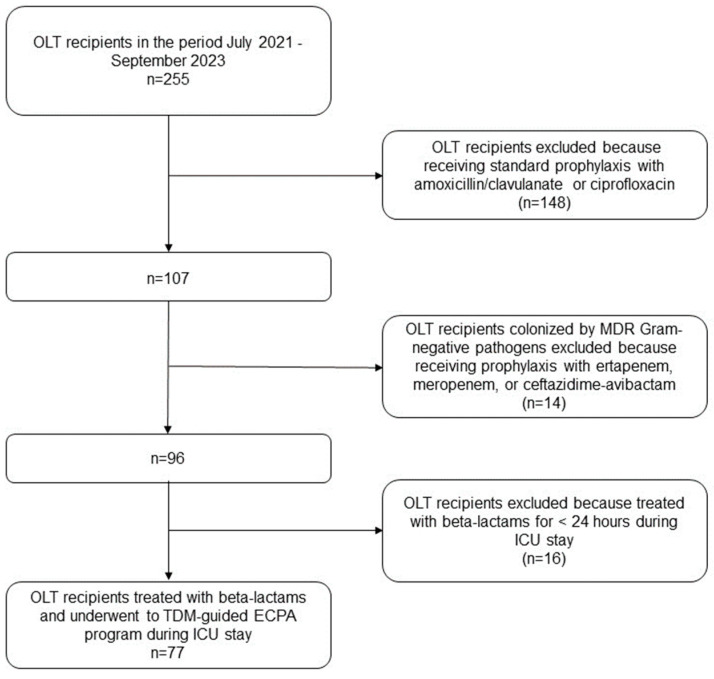
Flowchart of patient inclusion and exclusion criteria. ECPA: expert clinical pharmacological advice; ICU: intensive care unit; MDR: multidrug-resistant; OLT: orthotopic liver transplant; TDM: therapeutic drug monitoring.

**Figure 2 antibiotics-12-01599-f002:**
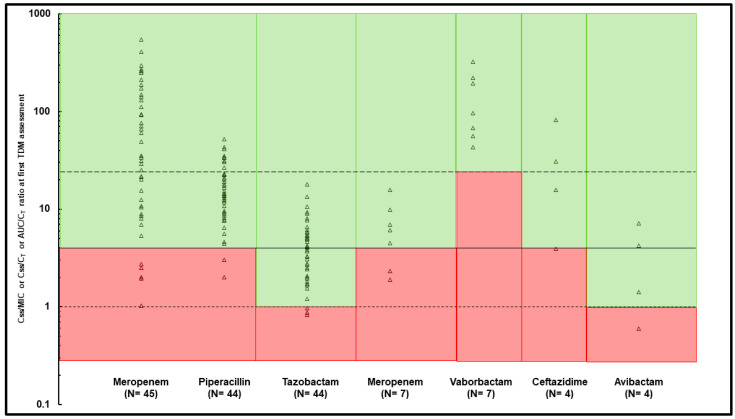
Early PK/PD target attainment assessed at first TDM assessment for each beta-lactam treatment course. Continuous line represents optimal *f*C_ss_/MIC ratio > 4 for meropenem, piperacillin, and ceftazidime; dotted lines represent optimal *f*C_ss_/C_T_ ratio > 1 for tazobactam and avibactam, and *f*AUC/C_T_ ratio > 24 for vaborbactam. Green area: optimal PK/PD target attainment, red area: quasi-optimal/suboptimal PK/PD target attainment.

**Table 1 antibiotics-12-01599-t001:** Demographics and clinical characteristics of included OLT recipients.

Demographics and Clinical Variables	OLT Recipients (*n* = 77)
*Patient demographics*	
Age (years) [median (IQR)]	57 (51–63)
Gender (male/female) [*n* (%)]	49/28 (63.6/36.4)
Body weight (Kg) [median (IQR)]	71 (60–81)
Body mass index (Kg/m^2^) [median (IQR)]	25.7 (22.5–27.8)
*Underlying liver diseases* [*n* (%)]	
Primary sclerosing cholangitis	12 (15.5)
Alcoholic + dysmetabolic cirrhosis	9 (11.7)
Alcoholic cirrhosis	7 (9.1)
HCV + hepatocarcinoma	7 (9.1)
Polycystic disease	6 (7.8)
Cholangiocarcinoma	5 (6.5)
Cryptogenic cirrhosis	5 (6.5)
Primary non-function	4 (5.2)
Congenital atresia biliary tract	3 (3.9)
HBV + alcoholic + dysmetabolic cirrhosis	3 (3.9)
HBV + HDV	3 (3.9)
HCV + alcoholic cirrhosis + hepatocarcinoma	3 (3.9)
Autoimmune hepatitis	2 (2.6)
Alcoholic cirrhosis + hepatocarcinoma	1 (1.3)
HBV	1 (1.3)
HBV + dysmetabolic cirrhosis	1 (1.3)
HBV + hepatocarcinoma	1 (1.3)
HCV	1 (1.3)
HCV + dysmetabolic	1 (1.3)
HCV + alcoholic cirrhosis	1 (1.3)
Chronic rejection	1 (1.3)
*OLT features*	
MELD score at transplantation [median (IQR)]	17 (11–29)
Donation after circulatory death [*n* (%)]	7 (9.1)
Re-OLT	5 (6.5)
*Severity of clinical conditions*	
SOFA score at ICU admission [median (IQR)]	6.5 (3.75–9.25)
Mechanical ventilation > 48 h [*n* (%)]	36 (46.8)
Vasopressors [*n* (%)]	50 (64.9)
Baseline CL_CR_ (mL/min/1.73 m^2^) [median (IQR)]	61 (32–109)
Continuous renal replacement therapy [*n* (%)]	28 (36.4)
Augmented renal clearance [*n* (%)]	15 (19.5)
*Outcome*	
ICU mortality	7 (9.1)

CL_CR_: creatinine clearance; ICU: intensive care unit; IQR: interquartile range; MELD: Model for End-Stage Liver Disease; OLT: orthotopic liver transplant; SOFA: sequential organ failure assessment.

**Table 2 antibiotics-12-01599-t002:** Features of beta-lactam treatment course implemented in the 77 OLT recipients during ICU stay.

Variables	Beta-Lactam Treatment Course (*n* = 100)
*Antimicrobial treatment* [*n* (%)]	
Empirical	57 (57.0)
Targeted	43 (43.0)
*Beta-lactam antimicrobials* [*n* (%)]	
Meropenem	45 (45.0)
Piperacillin-Tazobactam	44 (44.0)
Meropenem-Vaborbactam	7 (7.0)
Ceftazidime-Avibactam	4 (4.0)
*Site of infection* ^a^ [*n* (%)]	
VAP	17 (39.5)
cIAI	11 (25.6)
BSI	9 (20.9)
cIAI + BSI	4 (9.3)
VAP + BSI	2 (4.7)
*Gram-negative clinical isolates* ^b^ [*n* (%)]	
*Klebsiella pneumoniae*	16 (31.3)
*Enterobacter cloacae*	8 (15.7)
*Escherichia coli*	7 (13.7)
*Pseudomonas aeruginosa*	7 (13.7)
*Serratia marcescens*	3 (5.9)
*Acinetobacter baumannii*	3 (5.9)
*Klebsiella aerogenes*	2 (3.9)
*Hafnia alvei*	2 (3.9)
*Proteus mirabilis*	1 (2.0)
*Klebsiella oxytoca*	1 (2.0)
*Klebsiella ornithinolytica*	1 (2.0)
*TDM-based ECPA*	
Overall TDM-based ECPAs	245
*N* of TDM-based ECPA per treatment course [median (IQR)]	2 (1–3)
*N* of dosage confirmations at first TDM assessment [*n* (%)]	24 (24.0)
*N* of dosage increases at first TDM assessment [*n* (%)]	7 (7.0)
*N* of dosage decreases at first TDM assessment [*n* (%)]	69 (69.0)
Overall *n* of dosage confirmations [*n* (%)]	111 (45.3)
Overall *n* of dosage increases [*n* (%)]	15 (6.1)
Overall *n* of dosage decreases [*n* (%)]	119 (48.6)

BSI: bloodstream infection; cIAI: complicated intrabdominal infection; ECPA: expert clinical pharmacological advice; ICU: intensive care unit; TDM: therapeutic drug monitoring; VAP: ventilator-associated pneumonia. ^a^ Only the 43 targeted beta-lactam treatments were considered, ^b^ Overall, 51 different Gram-negative pathogens were identified for the 43 targeted beta-lactam therapies.

**Table 3 antibiotics-12-01599-t003:** PK/PD target attainments of CI beta-lactams in OLT recipients.

Beta-Lactam	Treatment Course
*Meropenem*	*45*
Daily dose (mg) [median (IQR)]	500 mg q6 h (500 mg q6 h–1000 mg q6 h)
*f*C_ss_ (mg/L) [median (IQR)]	13.7 (8.6–25.9)
*f*C_ss_/MIC ratio [median (IQR)]	58.8 (11.8–92.3)
Overall ECPAs	141
Overall *f*C_ss_/MIC ratio > 4 [optimal; *n* (%)]	126 (89.4)
Overall *f*C_ss_/MIC ratio 1–4 [quasi-optimal; *n* (%)]	12 (8.5)
Overall *f*C_ss_/MIC ratio < 1 [suboptimal; *n* (%)]	3 (2.1)
*f*C_ss_/MIC ratio > 4 at first TDM assessment [optimal; *n* (%)]	40 (88.9)
*f*C_ss_/MIC ratio 1–4 at first TDM assessment [quasi-optimal; *n* (%)]	5 (11.1)
*f*C_ss_/MIC ratio < 1 at first TDM assessment [suboptimal; *n* (%)]	0 (0.0)
*Piperacillin-tazobactam*	*44*
Daily dose (mg) [median (IQR)]	13,500 mg (9000 mg–18,000 mg)
Piperacillin *f*C_ss_ (mg/L) [median (IQR)]	86.4 (53.9–123.6)
Tazobactam *f*C_ss_ (mg/L) [median (IQR)]	10.5 (6.2–16.2)
Piperacillin *f*C_ss_/MIC ratio [median (IQR)]	10.9 (7.1–16.0)
Tazobactam *f*C_ss_/C_T_ ratio [median (IQR)]	2.6 (1.6–4.0)
Overall ECPAs	79
Overall optimal joint PK/PD target [*n* (%)]	69 (87.3)
Overall quasi-optimal joint PK/PD target [*n* (%)]	4 (5.1)
Overall suboptimal joint PK/PD target [*n* (%)]	6 (7.6)
Optimal joint PK/PD target at first TDM assessment [n (%)]	40 (90.9)
Quasi-optimal joint PK/PD target at first TDM assessment [n (%)]	1 (2.3)
Suboptimal joint PK/PD target at first TDM assessment [n (%)]	3 (6.8)
*Meropenem-vaborbactam*	*7*
Daily dose (mg) [median (IQR)]	2000 mg/2000 mg q8 h (1000 mg/1000 mg q8 h–2000 mg/2000 mg q8 h)
Meropenem *f*C_ss_ (mg/L) [median (IQR)]	30.0 (17.4–43.4)
Vaborbactam *f*C_ss_ (mg/L) [median (IQR)]	43.7 (29.7–51.9)
Meropenem *f*C_ss_/MIC ratio [median (IQR)]	4.4 (2.2–6.4)
Vaborbactam *f*AUC/C_T_ ratio [median (IQR)]	131.0 (60.8–143.9)
Overall ECPAs	11
Overall optimal joint PK/PD target [*n* (%)]	6 (54.5)
Overall quasi-optimal joint PK/PD target [*n* (%)]	5 (45.5)
Overall suboptimal joint PK/PD target [*n* (%)]	0 (0.0)
Optimal joint PK/PD target at first TDM assessment [*n* (%)]	5 (71.4)
Quasi-optimal joint PK/PD target at first TDM assessment [*n* (%)]	2 (28.6)
Suboptimal joint PK/PD target at first TDM assessment [*n* (%)]	0 (0.0)
*Ceftazidime-avibactam*	*4*
Daily dose (mg) [median (IQR)]	1250 mg q8 h (1250 mg q8 h–2500 mg q8 h)
Ceftazidime *f*C_ss_ (mg/L) [median (IQR)]	56.6 (28.8–75.0)
Avibactam *f*C_ss_ (mg/L) [median (IQR)]	12.8 (5.0–20.5)
Ceftazidime *f*C_ss_/MIC ratio [median (IQR)]	28.3 (14.4–37.5)
Avibactam *f*C_ss_/C_T_ ratio [median (IQR)]	3.2 (1.3–5.1)
Overall ECPAs	14
Overall optimal joint PK/PD target [*n* (%)]	13 (92.9)
Overall quasi-optimal joint PK/PD target [*n* (%)]	0 (0.0)
Overall suboptimal joint PK/PD target [*n* (%)]	1 (7.1)
Optimal joint PK/PD target at first TDM assessment [*n* (%)]	3 (75.0)
Quasi-optimal joint PK/PD target at first TDM assessment [*n* (%)]	0 (0.0)
Suboptimal joint PK/PD target at first TDM assessment [*n* (%)]	1 (25.0)

C_T_: target concentration; ECPA: expert clinical pharmacological advice; *f*AUC: free area under concentration-to-time curve; *f*C_ss_: free steady-state concentrations; IQR: interquartile range; MIC: minimum inhibitory concentration; OLT: orthotopic liver transplant; PK/PD: pharmacokinetic/pharmacodynamic; TDM: therapeutic drug monitoring.

**Table 4 antibiotics-12-01599-t004:** Univariate and multivariate analysis comparing beta-lactam treatment courses in which an early optimal vs. quasi-optimal/suboptimal PK/PD target was attained.

Variables	Early Optimal PK/PD Target Attainment (*n* = 88)	Early Quasi-Optimal/ Suboptimal PK/PD Target Attainment (*n* = 12)	Univariate Analysis *p* Value	Multivariate Analysis (OR; 95%CI)	Multivariate Analysis *p* Value
Age (years) [median (IQR)]	58 (53–64)	48.5 (33.5–60)	0.06	0.94 (0.88–1.01)	0.07
Gender (male/female) [*n* (%)]	53/35 (60.2/39.8)	12/0 (100.0/0.0)	0.007	−	−
Body weight (Kg) [median (IQR)]	72 (59–82.5)	72 (61.5–76.3)	0.95		
Body mass index (Kg/m^2^) [median (IQR)]	26.2 (23.1–28.4)	24.3 (21.8–27.0)	0.49		
MELD score at transplantation [median (IQR)]	21 (11.75–29)	14 (12.5–24)	0.54		
Donation after circulatory death [*n* (%)]	14 (15.9)	1 (8.3)	0.69		
SOFA score at ICU admission [median (IQR)]	8 (4–11)	4 (3–7)	0.35		
Mechanical ventilation > 48 h [*n* (%)]	48 (54.5)	6 (50.0)	0.77		
Vasopressors [*n* (%)]	61 (69.3)	8 (66.7)	0.99		
Continuous renal replacement therapy [*n* (%)]	43 (48.9)	4 (33.3)	0.37		
Augmented renal clearance [*n* (%)]	6 (6.8)	4 (33.3)	0.02	7.64 (1.32–44.13)	0.023
Empirical treatment [*n* (%)]	62 (70.5)	7 (58.3)	0.51		
Targeted treatment [*n* (%)]	26 (29.5)	5 (41.7)	0.51		
MIC value > EUCAST clinical breakpoint [*n* (%)]	1 (1.1)	4 (33.3)	<0.001	91.55 (7.12–1177.12)	<0.001

ICU: intensive care unit; IQR: interquartile range; MELD: Model for End-Stage Liver Disease; OR: odds ratio; PK/PD: pharmacokinetic/pharmacodynamic; SOFA: sequential organ failure assessment.

## Data Availability

The data presented in this study are available upon request from the corresponding author. The data are not publicly available due to privacy concerns.

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
