# Peer review of "Role of a Real-Time TDM-Based Expert Clinical Pharmacological Advice Program in Optimizing the Early Pharmacokinetic/Pharmacodynamic Target Attainment of Continuous Infusion Beta-Lactams among Orthotopic Liver Transplant Recipients with Documented or Suspected Gram-Negative Infections"

_antibiotics, 2023, doi:10.3390/antibiotics12111599_

Round 1
Reviewer 1 Report
Comments and Suggestions for Authors
I thought this article titled "Role of a real-time TDM-based expert clinical pharmacological advice program in optimizing early pharmacokinetic/pharmacodynamic target attainment of continuous infusion beta-lactams among orthotopic liver transplant recipients with documented or suspected Gram-negative infections" was very interesting.
My comment is below.
1.
Studies using PK/PD parameters should be evaluated using free drug concentration.
In authors article, it should be shown that the concentration of educts is measured, or that the total concentration is corrected by the protein binding rate.
2.
Meropenem is unstable after dissolution.
I think it is necessary to describe the effects of continuous infusion.
3.
Although the patients were suffering from various infectious diseases, the effects on pharmacokinetics were not described.
I think it should be written.
Author Response
RESPONSE TO REVIEWERS
Manuscript ID: antibiotics-2676763 entitled “Role of a real-time TDM-based expert clinical pharmacological advice program in optimizing early pharmacokinetic/pharmacodynamic target attainment of continuous infusion beta-lactams among orthotopic liver transplant recipients with documented or suspected Gram-negative infections” by Gatti et al.
Dear Editor,
We would like to thank you for the opportunity to resubmit a revised version of this manuscript. We appreciated the reviewers’ constructive comments. All have been carefully considered and incorporated, where and whenever possible, in the revision.
Our point-by-point responses are provided below.
Q= QUERY; A= ANSWER
Reviewer #1
I thought this article titled "Role of a real-time TDM-based expert clinical pharmacological advice program in optimizing early pharmacokinetic/pharmacodynamic target attainment of continuous infusion beta-lactams among orthotopic liver transplant recipients with documented or suspected Gram-negative infections" was very interesting.
A. We thank the reviewer for appreciating our manuscript.
My comment is below.
Q1. Studies using PK/PD parameters should be evaluated using free drug concentration. In authors article, it should be shown that the concentration of educts is measured, or that the total concentration is corrected by the protein binding rate.
A1. We thank the reviewer for this relevant comment. We agree with the fact that beta-lactam PK/PD targets should be evaluated by using free drug concentrations. Consequently, we calculated free beta-lactams concentrations according to the protein binding rate reported in literature (refer to Methods section – Line 346-349), and accordingly we adopted the fCss/MIC, the fCss/CT, and/or the fAUC/CT ratios as the best PK/PD targets for the different beta-lactams and beta-lactamase inhibitors. Furthermore, we modified accordingly the results (refer to Line 151-153 and Table 3), and we reported in the Discussion section among limitations that free beta-lactams concentrations were calculated according to data reported in literature (refer to Line 238-239).
Q2. Meropenem is unstable after dissolution. I think it is necessary to describe the effects of continuous infusion.
A2. We thank the reviewer for this important comment, allowing us to better clarify this issue. Several studies showed that the stability of meropenem, both as brand and generic formulations, is granted in aqueous solution for up to 8 hours (refer to references 52-54). Accordingly, continuous infusion of meropenem was granted by reconstituting aqueous solution every 6-8 hours and administering over 6-8 hours infusion, as reported in Methods section (refer to Line 297-299).
Q3. Although the patients were suffering from various infectious diseases, the effects on pharmacokinetics were not described. I think it should be written.
A3. We thank the reviewer for this comment. We added some statements about this issue (refer to Line 195-199).
Reviewer 2 Report
Comments and Suggestions for Authors
Your manuscript titled Role of a real-time TDM-based expert clinical pharmacological advice program in optimizing early pharmacokinetic/pharmacodynamic target attainment of continuous infusion beta-lactams among orthotopic liver transplant recipients with documented or suspected Gram-negative infections was well written. The study design is sound and data is presented in a clear and easy-to-understand fashion. The data and results support the conclusion.
Author Response
Reviewer #2
Your manuscript titled Role of a real-time TDM-based expert clinical pharmacological advice program in optimizing early pharmacokinetic/pharmacodynamic target attainment of continuous infusion beta-lactams among orthotopic liver transplant recipients with documented or suspected Gram-negative infections was well written. The study design is sound and data is presented in a clear and easy-to-understand fashion. The data and results support the conclusion.
A. We thank the reviewer for appreciating our manuscript.
Reviewer 3 Report
Comments and Suggestions for Authors
The study design and data collection are good. During the collection of clinical features of the patients, any probiotic supplement information collected from the patients? If collected, included that information also.
Author Response
Reviewer #3
The study design and data collection are good. During the collection of clinical features of the patients, any probiotic supplement information collected from the patients? If collected, included that information also.
A. We thank the reviewer for appreciating our manuscript and for the interesting question. However, according to routine clinical practice implemented at our Institution for the management of OLT recipients, none of the included patients in our study received probiotic supplements, as other approaches are usually implemented for minimizing the post-surgical risk of infections (i.e., early beginning of enteral nutrition). We added this information in the Results section (refer to Line 116-117).
Round 2
Reviewer 1 Report
Comments and Suggestions for Authors
Thank you for responding and responding constructively to my comments.
The manuscript has been improved now.